# Analysis of Influencing Factors for Stackable and Expandable Acoustic Metamaterial with Multiple Tortuous Channels

**DOI:** 10.3390/ma16206643

**Published:** 2023-10-11

**Authors:** Shaohua Bi, Fei Yang, Xinmin Shen, Jiaojiao Zhang, Xiaocui Yang, Heng Zhang, Wenqiang Peng

**Affiliations:** 1Field Engineering College, Army Engineering University of PLA, Nanjing 210007, China; 17337434454@163.com (S.B.); 19962061916@163.com (F.Y.); zhangheng4216@sina.com (H.Z.); 2Engineering Training Center, Nanjing Vocational University of Industry Technology, Nanjing 210023, China; 2019101052@niit.edu.cn; 3College of Aerospace Science and Engineering, National University of Defense Technology, Changsha 410073, China; plxhaz@126.com

**Keywords:** stackable and expandable acoustic metamaterial, multiple tortuous channels, sound absorption performance, influencing factors, acoustic finite element simulation, noise reduction, experimental validation, sound absorption mechanism

## Abstract

To reduce the noise generated by large mechanical equipment, a stackable and expandable acoustic metamaterial with multiple tortuous channels (SEAM–MTCs) was developed in this study. The proposed SEAM–MTCs consisted of odd panels, even panels, chambers, and a final closing plate, and these component parts could be fabricated separately and then assembled. The influencing factors, including the number of layers *N*, the thickness of panel *t*_0_, the size of square aperture *a*, and the depth of chamber *T*_0_ were investigated using acoustic finite element simulation. The sound absorption mechanism was exhibited by the distributions of the total acoustic energy density at the resonance frequencies. The number of resonance frequencies increased from 13 to 31 with the number of layers *N* increasing from 2 to 6, and the average sound absorption coefficients in [200 Hz, 6000 Hz] was improved from 0.5169 to 0.6160. The experimental validation of actual sound absorption coefficients in [200 Hz, 1600 Hz] showed excellent consistency with simulation data, which proved the accuracy of the finite element simulation model and the reliability of the analysis of influencing factors. The proposed SEAM–MTCs has great potential in the field of equipment noise reduction.

## 1. Introduction

The harm of noise generated by large mechanical equipment increases along with the improvement of its working power, which seriously affects the surrounding managers and workers [1,2,3,4,5,6]. For example, Sarmadi et al. studied noise control in small power plants, attempting to reduce the noise pollution using copper and nickel alloy foam. Similarly, a mathematical model for determining an optimal noise barrier arrangement was developed by Choi et al. [2], which was expected to be helpful both to construction companies and to people near construction sites. Meanwhile, Farooqi et al. [3] investigated the noise pollution to local residents in Faisalabad due to the development of the robust industrial and transport systems, including both the auditory and nonauditory effects. Analogously, the noise produced by the various machines and equipment used in the production processes in Nigerian manufacturing companies was analyzed by Bolaji et al. [4], the results of which indicated that the effects of noise on workers were more physiological than psychological. Moreover, Chivu et al. [5] analyzed the noise generated in a production unit where filling of polyurethane foam tubes was performed, the objective of which was to determine the level of noise in the factory and how it acted as a physical and professional risk factor, including its effects on the human body. Furthermore, a novel methodology for the prediction, evaluation, and analysis of these industrial noise sources was proposed by del Amor et al. [6], and a new tool for predicting and categorizing outdoor noise from its measurement at the source was developed. It can be found that the prevention of noise pollution is important in modern society [1,2,3,4,5,6], which has attracted research interests in the crossing fields of materialogy and environics.

Thus, many sound-absorbing materials or structures have been developed to control noise, such as microperforated panels [7,8], porous materials [9,10], acoustic metamaterials [11,12], etc. For instance, Li et al. [7] proposed microperforated composite sound absorption structures to reduce noise in helicopter cabins, which realized an amplitude of more than 20 dB in 500–2000 Hz range. Similarly, multilayer microperforated panels with no more than four layers were optimized by Yang et al. [8], and the optimal average sound absorbing coefficients in the 100–6000 Hz range were 0.5721, 0.6629, 0.6833, and 0.6936. In addition, Tang and Yan [9] summarized the advances concerning the acoustic absorption of various fibrous materials, including inorganic fibers, metallic fibers, synthetic fibers, natural fibers, and nanofibrous membranes. Moreover, the conditions to observe perfect sound absorption by rigidly backed layers of rigid-frame high-porous materials were proposed by Jiménez et al. [10], in which a single layer of highly porous material, a layer of highly porous material with an air gap, and an optimized multilayer structure were analyzed. Pavan and Singh [11] presented a novel porous labyrinthine acoustic metamaterial containing a folded slit labyrinthine structure in a microporous matrix, and near-perfect sound absorption at the low-frequency range of 200–500 Hz for various compositions was proven in theory, simulations, and experiments. Furthermore, to achieve multiple absorption peaks at given low-frequency targets for a substation noise source, an acoustic multi–layer Helmholtz resonance metamaterial was developed by Duan et al. [12], and two groups of resonance peaks were generated at 100 Hz and 400 Hz when the thickness was only 1/30th of the working wavelength. These sound absorbing materials and structures [7,8,9,10,11,12,13] have made great contributions in the field of noise control, which provide effective guidance and important references for developing more novel sound absorbers.

Among the present sound-absorbing materials or structures [7,8,9,10,11,12,13], acoustic metamaterials have significant advantages in controlling sound waves in the low-frequency range, and optimizing their structures can enable functionality based on new physical phenomena [14,15,16,17,18,19,20]. For example, Gao et al. [14] summarized the basic classification, underlying physical mechanism, application scenarios, and emerging research trends for both passive and active noise-reduction metamaterials. Similarly, a new layered membrane metamaterial was developed by Ciaburro et al. [15] based on the three layers of a reused PVC membrane with reused metal washers attached, which behaved like an acoustic absorber even at low frequencies. Analogously, Ciaburro and Iannace [16] proposed a new membrane-type acoustic metamaterial using a recycled cork membrane and affixing to this membrane masses obtained by reusing thumbtacks and buttons, which could be favorable in terms of the efficient use of energy resources and raw materials. Meanwhile, a similar fractal coiled acoustic metamaterial was designed by Cui et al. [17] for low-frequency noise control by combining a perforated plate and a coiled back cavity structure, and its sound energy dissipation mechanism was revealed by finite element analysis. In addition, Naimusin and Janusevicius [18] studied structures created from metamaterial with plastic for their sound-absorbing properties, which gained a good absorption peak at 315 Hz of 0.94 using a combined 100 mm long resonator. Moreover, acoustic metamaterials based on Helmholtz resonators and capable of attenuating sound up to 30 dB were developed by Casarini et al. [19], which could be applied for noise control in small-scale electroacoustic devices and sensors. Furthermore, Sharafkhani [20] converted a single-band Helmholtz resonator–based sound absorber into a multi-band absorber while maintaining its thickness at 55.1 mm, and perfect absorption was realized for the main components of power transformer noise. Therefore, acoustic metamaterials have been considered as the most promising sound absorbers for noise reduction [14,15,16,17,18,19,20].

However, there are two major problems that limit the practical application of most acoustic metamaterials. The first is how to realize mass production. Most of the acoustic metamaterials are fabricated by additive manufacturing, which is more suitable for small batch customization instead of flow line production. The second is that the sound absorption performance of a certain acoustic metamaterial is established when it is produced; one cannot easily adjust its parameters to adapt to the changes in the noise environment. To overcome these two problems, a stackable and expandable acoustic metamaterial with multiple tortuous channels (SEAM–MTCs) is proposed in this research, which consists of several panel layers and chamber layers. Its structures and operating principle are presented firstly, which show its advantages in rapid manufacturing and convenient adjustment. Then, its sound absorption properties are qualitatively investigated by the theoretical modeling [21,22] and quantitatively analyzed by finite element simulation [23,24]. Later, we study the effects of influencing factors on the sound absorption performance of the proposed SEAM–MTCs, which consist of the number of layers, the thickness of the panel, the size of the square aperture, and the depth of the chamber. After that, the sound absorption mechanism is further revealed by the distributions of the total acoustic energy densities at the resonance frequencies [25], and the sound absorption principle is summarized as well. Finally, taking four-layer SEAM–MTCs as an example, the sample is assembled, and its sound absorption coefficients are tested according to the transfer function method [26,27,28], which demonstrates the effectiveness of this novel acoustic metamaterial and the reliability of acoustic finite element simulation. Through the explanation of structural composition, the investigation of influencing factors, the study of absorption performance, the analysis of the absorption mechanism, and the testing of the experimental sample, the major advantages of this proposed SEAM–MTC are demonstrated, which mainly include a wide absorption band, fine machinability, suitability for mass production, easy assembly, convenient adjustment, etc.

## 2. Materials and Design

Taking the SEAM–MTCs with 4 layers as an example, its schematic diagram is shown in Figure 1. The whole structure in Figure 1a consists of 2 odd panels, 2 even panels, 4 chambers, and 1 final closing plate, which can be seen in the expanded view in Figure 1b. The square apertures in the panels and the cavities in the chambers together formed the channel. When the length of the equivalent channel reached kλ/4 (k = 1, 2, 3, …; λ is the wave length of the incident sound wave), it could realize excellent sound absorption effects at this frequency point. Generally speaking, the major influencing factors for the SEAM–MTCs were the size of the panel and that of chamber, which consisted of the number of layers *N*, the thickness of the panel *t*_0_, the length of the side for the square aperture on the panel *a*, and the length of the chamber *T*_0_, as shown in Figure 1c–e. There were 44 channels in total, and they were equally divided into 11 groups. For the 4 single channels in each group, their lengths were equal. In order to facilitate the differentiation in the later study, the 11 groups were labelled as C01 to C11, with the length of chamber from large to small, which can be seen in the summarized parameters in Table 1 for the 11 groups of tortuous channels in the SEAM–MTCs. Taking the following experimental validation into consideration, the size of the proposed acoustic metamaterial was limited to 70 mm, which ensured the sample fit into a cylindrical tube with the diameter of 100 mm; the size of the standing wave tube utilized in this research was Ф100 mm [28]. Thus, the thickness of the side wall among the different chambers was maintained at 2 mm, as shown in Figure 1f. Meanwhile, except for the C11group of channels, the width of the chambers for the other 10 groups of channels were maintained at 5 mm, which is shown in Figure 1e as well. For the C11group of channels, the length of side of the square aperture was 4 mm, and the width of the chamber was 4 mm as well, which meant that all were straight cavities. To make the distributions of length of tortuous channels as uniform as possible, the length of the single chamber for the 10 groups of channels from C01 to C10 was selected as 20 mm, 18 mm, 17 mm, 16 mm, 14 mm, 13 mm, 12 mm, 11 mm, 10 mm and 9 mm, respectively, as shown in Figure 1f and Table 1. These values were tunable as required, but they should meet the following conditions: the sum of lengths of C01 and C10 was kept as 29 mm; that of lengths of C02, C07, and C08 was kept as 41 mm; that of lengths of C04, C05, C06, and C09 was kept as 53 mm. It could be found that this acoustic metamaterial was stackable in the depth direction and expandable in the plane direction, and the panels and chambers could be fabricated separately and then assembled, which was beneficial to markedly reduce the fabrication costs and to allow flexibility in adjusting the sound absorption performance. In this way, the proposed SEAM–MTCs could be mass-produced easily.

## 3. Theoretical Modeling and Finite Element Simulation

### 3.1. Theoretical Modeling

The theoretical model for the SEAM–MTCs was constructed according to the Fabry–Pérot resonance principle [29,30], and the corresponding theoretical sound absorption coefficient could be calculated based on the acoustic impedance. Each group of tortuous channels with the same parameters could be considered as a set of Fabry–Pérot resonators [29,30], and their length *L_n_* could be roughly calculated using Equation (1). Here, *N*, *t*_0_, *T*_0_, *w*, *l_n_*, and *a* are the number of layers, the thickness of panel, the depth of chamber, the width of chamber, the length of chamber, and the length of the side of the square aperture respectively, which are consistent with the marked parameters in Figure 2, Figure 3 and Figure 4. The cuboidal chamber was equivalent to the channel with a sectional area equivalent to that of the square aperture, which was convenient for the following calculation.
(1)$$L_{n}=N\times \left(t_{0}+\frac{\sqrt{T_{0}^{2}+l_{n}^{2}}\times w}{a}\right)$$

The acoustic impedance *Z_n_* for each group of tortuous channels could be derived using Equation (2). *Z_n_*_0_ was the acoustic impedance of a single tortuous channel, which could be calculated by Equation (3). *σ_n_* was the perforation ratio, which was obtained using Equation (4), and *A* was the length of a side for the square metamaterial cell.
(2)$$Z_{n}=\frac{Z_{n0}}{\sigma_{n}}$$
(3)$$Z_{n0}=-iZ_{cn}\cot\left(k_{cn}L_{n}\right)$$
(4)$$\sigma_{n}=4\times \frac{a^{2}}{A^{2}}$$

In Equation (3), *Z_cn_* is the effective characteristic impedance of air in the tortuous channel, which is determined using Equation (5); *k_cn_* is the effective transfer function of air in the tortuous channel, which can be determined using Equation (6).
(5)$$Z_{cn}=\sqrt{\frac{\rho_{cn}}{C_{cn}}}$$
(6)$$k_{cn}=\sqrt{\rho_{cn}C_{cn}}$$

In Equations (5) and (6), $\rho_{cn}$ and $C_{cn}$ are the effective density and the effective volumetric compressibility of air, which can be calculated using Equations (7) and (8) according to the thermal viscosity acoustic theory [31,32,33].
(7)$$\rho_{cn}=\rho_{0}\frac{va^{4}}{4i\omega }{\left\{\sum_{m=0}^{\infty }\sum_{n=0}^{\infty }{\left[\alpha_{m}^{2}\beta_{n}^{2}\left(\alpha_{m}^{2}+\beta_{n}^{2}+\frac{i\omega }{v}\right)\right]}^{-1}\right\}}^{-1}$$
(8)$$C_{cn}=\frac{1}{P_{0}}\left\{1-\frac{4i\omega \left(\gamma -1\right)}{v^{\prime }a^{4}}{\sum_{m=0}^{\infty }\sum_{n=0}^{\infty }\left[\alpha_{m}^{2}\beta_{n}^{2}\left(\alpha_{m}^{2}+\beta_{n}^{2}+\frac{i\omega \gamma }{v^{\prime }}\right)\right]}^{-1}\right\}$$

In Equations (7) and (8), $\rho_{0}$ is the density of air under normal temperature and standard atmospheric pressure, 1.225 Kg/m^3^; v is the kinematic viscosity of air, which can be determined using Equation (9); ω is the angular frequency of a sound wave, which is derived using Equation (10); $\alpha_{m}$ and $\beta_{n}$ are the intermediate computation constants, which can be determined using Equations (11) and (12); $P_{0}$ is the standard atmospheric pressure under normal temperature, 1.01325 × 10^5^ Pa; γ is the specific heat ratio, 1.4; $v^{\prime }$ can be calculated using Equation (13).
(9)$$v=\frac{\mu }{\rho_{0}}$$
(10)ω=2πf
(11)$$\alpha_{m}=\frac{\left(m+1/2\right)\pi }{a}$$
(12)$$\beta_{n}=\frac{\left(n+1/2\right)\pi }{a}$$
(13)$$v^{\prime }=\frac{\kappa }{\rho_{0}C_{v}}$$

In Equation (9), *μ* is the coefficient of kinetic viscosity, 1.8 × 10^–5^ Pa·s. In Equation (10), *f* is the frequency of a sound wave. In Equation (13), κ and $C_{v}$ are heat conductivity and specific heat capacity under the constant volume mode, respectively, and their values are 0.0258 W/(m·K) and 718 J/(Kg·K).

According to the classical acoustic–electric analogy method [34,35,36], the total acoustic impedance *Z_total_* of the SEAM–MTCs could be derived using Equation (14), and the corresponding theoretical sound absorption coefficient *α* could be calculated using Equation (15).
(14)$$Z_{total}=\frac{1}{\sum_{1}^{n}\left(1/Z_{n}\right)}$$
(15)$$\alpha =1-{\left|\frac{Z_{total}-\rho_{0}c_{0}}{Z_{total}+\rho_{0}c_{0}}\right|}^{2}$$

Based on the Fabry–Pérot resonant principle [29,30], the theoretical sound absorption performance of each group of tortuous channels was investigated. However, it could be judged from the previous theoretical modeling process that there were many assumptions, approximations, equivalences, and omissions, which indicated that the accuracy of theoretical model was low [37,38]. Therefore, the acoustic finite element simulation method was selected to quantificationally analyze the sound absorption performance of the proposed acoustic metamaterial, and its reliability was validated by experimental tests as well.

### 3.2. Finite Element Simulation

The acoustic finite element simulation model of the proposed SEAM–MTCs for four layers was constructed in COMSOL multi-physics field simulation software 5.5, as shown in Figure 2. The perfect matching layer in Figure 2a was used to simulate the infinite air domain next to the acoustic field in the actual scene, which could fully absorb the incident sound waves without any reflected sound waves [39,40,41]. The background acoustic field was utilized to simulate the acoustic source, the type and propagation direction of which could be defined in combination with the actual engineering scene. The acoustic metamaterial in Figure 2a corresponds to the air domain in Figure 1a. The wall of the chamber and that of the aperture were solid materials, in which the dielectric density and propagation speed of sound waves were much larger than those in the air medium. The acoustic impedance of these walls was obviously larger than that of air, so the walls were regarded as a hard boundary, and the simulation model could only focus on the air domain inside the acoustic metamaterial.

The selected parameters in the acoustic finite element simulation process are summarized in Table 2 based on the thermal viscosity acoustics module [42,43]. Although the simulation accuracy could increase with a finer grid, the calculation time would also increase significantly. The selected parameters utilized in this study aimed to obtain the balance between simulation accuracy and calculation time. Taking into consideration the potential application scenarios of the proposed SEAM–MTCs, the investigated frequency range was set as 200–6000 Hz, with the uniform interval of 2 Hz. By adjusting the structural parameters of the proposed SEAM–MTCs in Figure 2, the effects of the influencing factors could be investigated, allowing examination of the sound absorption properties and revealing the sound absorption mechanism.

The streamline diagram of acoustic velocities in the group of tortuous channels with the same parameters is shown in Figure 3. It can be observed that the length of the motion path of the sound wave in Figure 3a is obviously larger than that in Figure 3b, which indicates that the former channels could achieve the resonance effect with incident sound wave at a low frequency, and the latter mainly absorbed the incident sound wave at the high-frequency region. Meanwhile, it can be found that the streamline diagram in the four single channels for each group is the same, which means that the sound absorption result is the coupling action of these four channels together. Otherwise, the sound absorption effect of only 1 single channel is limited, and the peak sound absorption coefficient at the resonance frequency is small. This is why the 44 channels in the proposed SEAM–MTCs were uniformly divided into 11 groups instead of 44 separate ones. Moreover, it could be judged from Figure 3 that the actual length of the active channel was majorly determined by the thickness of the panel and the depth and length of the chamber, which is consistent with the former analysis in the theoretical modeling process.

## 4. Parametric Analysis

The effects of four parameters were investigated, which included the number of layers *N*, the thickness of the panel *t*_0_, the size of the square aperture *a*, and the depth of the chamber *T*_0_. The parameter combination of *N* = 4, *t*_0_ = 2 mm, *a* = 5 mm, and *T*_0_ = 10 mm was treated as the anchoring group, and the selected ranges of values for *N*, *t*_0_, *a*, and *T*_0_ were (2, 4, 6), (2 mm, 3 mm, 4 mm), (3 mm, 4 mm, 5 mm), and (8 mm, 10 mm, 12 mm), respectively.

### 4.1. The Number of Layers N

The effects of number of layers *N* on the sound absorption performance of the SEAM–MTCs are shown in Figure 4, as obtained by the acoustic finite element simulation. It can be found that all the sound absorption peaks shifted to the low-frequency direction along with the increase of the number of layers, because the length of each tortuous channel increased accordingly. Meanwhile, it can be observed that the number of sound absorption peaks exceeded 11, although there were only 11 groups of tortuous channels with different parameters. The major reason for this phenomenon is that each group of tortuous channels could generate multiple sound absorption peaks instead of just one, because it could effectively absorb the incident sound wave when the length of the tortuous channel was 1/4 × kλ (k = 1, 2, 3, …; λ is the wave length of the incident sound wave), which could be determined using the theoretical model of the sound absorption coefficient for a single tortuous channel in Equation (1). However, the addition of each single layer indicated that the total thickness of the acoustic metamaterial increased *t*_0_ + *T*_0_, which meant the occupied space would become larger accordingly. Thus, to develop a practical SEAM–MTCs, its parameters should be selected to balance the sound absorption property and the occupied space.

### 4.2. The Thickness of Panel t_0_

Similarly, the effects of the thickness of panel *t*_0_ on the sound absorption performance of the proposed SEAM–MTCs are shown in Figure 5. It can be found that the sound absorption curve shifted to the low-frequency direction gradually with the increase of thickness of panel *t*_0_, because the equivalent length increased slightly for each tortuous channel. However, the variation of the corresponding sound absorption coefficients was relatively small, since the increase of the thickness of panel *t*_0_ had a small effect on the equivalent length of tortuous channel, which could be determined using Equation (1) as well. Moreover, the peak sound absorption coefficients for these resonance frequencies were also affected, so the adjustment of thickness of panel *t*_0_ could help to adjust the sound absorption performance for a certain frequency band while having little effect on the occupied space.

### 4.3. The Size of Square Aperture a

Analogously, the effects of the length of the side of the square aperture *a* on the sound absorption performance of the proposed SEAM–MTCs are shown in Figure 6. It can be found that along with the decrease of the length of the side of the square aperture *a*, the sound absorption curve shifted to the low-frequency direction, and the sound absorption performance deteriorated. The decrease of *a* increased the acoustic impedance of the square aperture, which would be favorable in obtaining the resonant absorbing effect at a lower frequency point. However, the total thickness of SEAM–MTCs had no change, so the coupling sound absorption effect was weakened for each group of channels, which could be judged from the decrease of peak sound absorption coefficient at each resonance frequency point, as seen in Figure 6. Therefore, variation of the length of the side of the square aperture *a* was not suitable for adjusting the sound absorption performance of the proposed SEAM–MTCs.

### 4.4. The Depth of Chamber T_0_

Homoplastically, the effects of the depth of chamber *T*_0_ on the sound absorption properties of the proposed SEAM–MTCs are shown in Figure 7. It can be observed that with the increase of depth of chamber *T*_0_, the sound absorption curve shifted to the low-frequency direction; the peak sound absorption coefficients at these resonance frequencies had little change, but the absorption bandwidth of a single peak decreased significantly, which was similar to the effect of the number of layers *N* in Figure 4.

Through the analysis of these influencing factors on SEAM–MTCs, it could be found that its sound absorption performance was adjustable through selecting the appropriate structure parameters, which is favorable with regard to promoting its practical application in the field of noise reduction.

## 5. Sound Absorption Mechanism

It could be judged from the effects of the parameters in Figure 4, Figure 5, Figure 6 and Figure 7 that the number of layers *N* was the most important influencing factor on the sound absorption performance of the proposed SEAM–MTCs. Therefore, the sound absorption mechanism of this acoustic metamaterial with various layers was investigated in order to reveal its sound absorption principle.

### 5.1. The Acoustic Metamaterial with Two Layers

When the number of layers for the proposed SEAM–MTCs was two, the peak frequency points on the sound absorption curve were determined, marked as green stars in Figure 8. It could be judged from Figure 1e that there were 11 groups of channels, which indicated that there would be 11 sound absorption peaks in each frequency band in theory. However, the final sound absorption effect of the SEAM–MTCs was not just a superposition of individual absorption peaks; some of the sound absorption results were the coupling effects of several groups of channels. The first frequency band was divided as [1470 Hz, 3258 Hz] roughly, and the second frequency band was from 4736 Hz to a frequency exceeding 6000 Hz. It could be found that all these resonance frequency points were in the high-frequency range, because the total thickness of SEAM–MTCs for two layers was *N* × (*t*_0_ + *T*_0_) + *t*_0_ = 26 mm, and the maximum equivalent length of the channel was 48.7 mm, which was approximately 1/4 of the wave length of the sound wave at the first resonance frequency of 1662 Hz. For the second frequency band, the first resonance frequency in the band was 4796 Hz, and the corresponding wave length was around 71.5 mm, which was closer to 1.5 times, rather than 2 times, the maximum length of the channel, 48.7 mm. The major reason for this phenomenon was that the calculated equivalent length of channel was roughly estimated, and the calculation accuracy had some relationship to the frequency of the incident sound wave. Along with the increase of frequency, the corpuscular property of the incident sound wave increased gradually, and its volatility decreased accordingly, which resulted in the actual length of each channel being smaller than the calculated results obtained by Equation (1).

The distributions of total acoustic energy density at these resonance frequencies are show in Figure 9, where the number of layers is two. It can be observed that each sound absorption peak was generated by single groups of channels or several groups of channels. The correspondence of the resonance frequency points to the 11 groups of channels for the two layers are summarized in Table 3, in which the marked “1” indicates that this group of channels made obvious contributions to the generation of the corresponding sound absorption peak. The distributions of total acoustic energy density are given in Figure 9. Meanwhile, it can be observed from Table 3 that most of the sound absorption peaks were generated by a single group of channels, because the variations of length between these neighboring groups of channels were large, which indicated that the coupling effects among the various channels with different lengths were weak and the sound absorption effects were mainly contributed by a certain group of channels. It should be noted that “these neighboring groups of channels” refers to channels with similar lengths instead of channels close in geographic space. For example, the neighboring groups of channels for C02 were C01 and C03 rather than the surrounding channels, C04, C05, C07, and C08. Moreover, it could be found that there were 10 sound absorption peaks in the first frequency band [1470 Hz, 3258 Hz], because the first resonance frequency point of 1662 Hz was mainly generated by the C02 group of channels with the assistance of the C01 group and that of the C03 group, which can be seen in Figure 9a. Furthermore, it can be observed from Figure 9n that although there were 13 sound absorption peaks, the values of the peak sound absorption coefficients at some resonance frequencies were small, especially for the resonance frequency point 4796 Hz in Figure 9k. Generally speaking, the sound absorption effect was better when the total acoustic energy density was larger in the group of resonant channels. It can be seen that the upper and lower limits of the legend in Figure 9k are minimum among the results in Figure 9, which means that the coupling sound absorption effect generated by this group of tortuous channels was weak. The major reason for this phenomenon was that this sound absorption peak was generated by the C01 group of tortuous channels with the maximum equivalent length, and it was difficult to realize fine sound absorption effects at the edge of each sound absorption band.

### 5.2. The Acoustic Metamaterial with Four Layers

Homoplastically, when the number of layers for the proposed SEAM–MTCs was four, the resonance frequency points on the sound absorption curve were labelled, as shown by the green stars in Figure 10. When compared with the results in Figure 8 for the acoustic metamaterial with two layers, there were more sound absorption peaks, as seen in Figure 10. The whole frequency range was divided into several frequency bands, similar to the relationship between the length of the channel and that of incident sound wave. The first frequency band was roughly divided as [740 Hz, 1718 Hz], and there were 10 sound absorption peaks in this band, which corresponded to the lengths of the tortuous channels close to 1/4 wavelength for these resonance frequencies. The second frequency band was set as [2550 Hz, 4480 Hz], with 10 sound absorption peaks in this band, which corresponded to the lengths of tortuous channels close to 1/2 wavelength for these resonance frequencies. Similarly, the third frequency band was from 4580 Hz to a frequency exceeding 6000 Hz, and there were four sound absorption peaks in [4580 Hz, 6000 Hz], which corresponded to the lengths of tortuous channels close to 3/4 wavelength for these resonance frequencies. It can be observed from Figure 10 that the sound absorption curve shifted to the low-frequency direction relative to the results for two layers in Figure 8.

The distributions of total acoustic energy density at these 24 resonance frequencies are shown in Figure 11, with the number of layers as four, and the correspondence of resonance frequency points to the 11 groups of channels is summarized in Table 4. It can be judged from Figure 11 that the average sound absorption effect at each resonance frequency point was enhanced, and all the peak sound absorption coefficients exceeded 0.65. The average sound absorption coefficient in [200 Hz, 6000 Hz] was improved from 0.5169 for the acoustic metamaterial with two layers to 0.5607 for the acoustic metamaterial with four layers. Especially for the low-frequency range [200 Hz, 1600 Hz], the average sound absorption coefficient increased from 0.2435 to 0.7198 with the increase of the number of layers from two to four, which exhibited a significant improvement in the low-frequency sound absorption performance. It was demonstrated that the addition of a layer was the most effective way to improve the sound absorption property, and the fine adjustment of the sound absorption frequency band could be realized by tuning the other parameters.

### 5.3. The Acoustic Metamaterial with Six Layers

Analogously, when the number of layers for the proposed SEAM–MTCs was six, the resonance frequency points on the sound absorption curve were marked, as shown by the green stars in Figure 12. Relative to the results for two layers in Figure 8 and those for four layers in Figure 10, the number of frequency bands and that of the sound absorption peaks further increased. It can be judged from Figure 12 that the four frequency bands were roughly divided as [490 Hz, 1158 Hz], [1732 Hz, 2748 Hz], [2848 Hz, 3854 Hz], and from 3976 Hz to a frequency exceeding 6000 Hz; the numbers of sound absorption peaks were 10, 8, 7, and 6, respectively. It can be observed from Figure 12 that the sound absorption curve further shifted to the low-frequency direction. Relative to the results for four layers as shown in Figure 10, the average sound absorption coefficient in the frequency range of [200 Hz, 6000 Hz] was further improved from 0.5607 to 0.6160. In particular, for the low-frequency range of [500 Hz, 1000 Hz], the average sound absorption coefficient of the SEAM–MTCs increased significantly, from 0.5752 to 0.8551, along with the increase of number of layers from four to six. The first resonance frequency in Figure 12 was 586 Hz, and its corresponding wavelength was approximate to 585 mm. According to Equation (1), the maximum length of channels for six layers was 146.1 mm; this was close to the 1/4 wavelength of the incident sound wave corresponding to the resonance frequency 586 Hz, which was consistent with the normal sound absorption principle of the Fabry–Pérot resonator [29,30]. Moreover, it could be observed that the spacing of the latter three frequency bands was small, and there existed some overlaps. This is because each group of tortuous channels in SEAM–MTCs could generate a series of resonance frequencies, and their values were mainly determined by their equivalent lengths, which indicated that the resonance frequencies might coincide or approach under certain conditions.

The distribution of total acoustic energy density at the 31 resonance frequencies is shown in Figure 13 for six layers, and the correspondence of the resonance frequency points to the 11 groups of tortuous channels is summarized in Table 5. It can be found that the principle to generate these resonance frequencies in the fourth frequency band was different from those of the previous three frequency bands, because the overlaps of resonance frequencies generated by different groups of tortuous channels were heavy and complicated. Based on Equation (1), the lengths of the tortuous channels for the 11 groups in the SEAM–MTCs for six layers were 146.1 mm, 135.5 mm, 130.3 mm, 125.2 mm, 115.2 mm, 110.4 mm, 105.7 mm, 101.2 mm, 96.9 mm, 92.7 mm, and 72 mm, respectively corresponding to the groups of tortuous channels from C01 to C11. When the multiples of the lengths of two channels met certain conditions, their resonance frequencies would overlap. Taking the resonance frequency 4026 Hz as an example, the corresponding wavelength was about 85.2 mm, and it was primary generated by the C01 and C06 groups of tortuous channels. It could be calculated that the lengths of the C01 group of channels were about 7/4 of the wavelength and those of C06 group were approximately 5/4 of the wavelength, so their resonant sound absorption effects overlapped approximatively at this resonance frequency point. Once more, taking the resonance frequency 4026 Hz as an example, the corresponding wavelength was about 58.1 mm, and it was primary generated by the C01, C04, C07, and C11 groups of channels. It could be derived that the lengths of C01, C04, C07, and C11 groups of channels were about 10/4, 9/4, 7/4, and 5/4 of the wavelength, respectively, which indicated that their resonant sound absorption effects overlapped approximatively at this resonance frequency point. Though these analyses were not absolutely accurate, they could qualitatively demonstrate the sound absorption principle and mechanism of the proposed SEAM–MTCs. It could be observed from Table 5 that the first frequency band was completely distinct from the second frequency band, while there existed some overlap at the resonance frequencies 2918 Hz, 2982 Hz, 3204 Hz and 3276 Hz between the second and third frequency bands. There was too much overlap between the third and fourth frequency bands, and the high sound absorption coefficients could be maintained when the frequency exceeded 5000 Hz, because these 11 groups of channels could contribute the resonant sound absorption effects densely with high efficiency in the high-frequency range. Therefore, along with the increase of number of layers, the sound absorption mechanism was the same, but the presented sound absorption principles were different in the high-frequency band; thus, both the sound absorption requirement and the occupied space should be into consideration in the analysis.

### 5.4. Acoustic Characteristic Parameters

In order to better reveal the sound absorption mechanism of SEAM–MTCs, the distributions of some acoustic characteristic parameters are shown in Figure 14, at the resonance frequency of 2586 Hz and with the number of layers at four; these include acoustic pressure, acoustic velocity, local acceleration, viscous power density, total acoustic energy density, and total specific entropy variation, corresponding to Figure 14a–f, respectively. For better display effect, only the C01 group of tortuous channels is shown in Figure 14 and the other groups of channels are hidden. It can be observed from Figure 14a that there existed differences in acoustic pressure between the tortuous channels and background acoustic field (the acoustic pressure is set as 1 Pa), which could result in the rapid motion of air in the aperture (as shown in Figure 14b); the variation of velocity was dramatic as well (as shown in Figure 14c). There were frictions between the moving air and the side wall of the aperture, which could generate the thermal viscous effect; this can be seen in the distribution of viscous power density in Figure 14d. The distribution of total acoustic energy density in Figure 14e and that of total specific entropy variation in Figure 14f further demonstrate the sound absorption effect.

## 6. Experimental Validation

As mentioned above, the proposed SEAM–MTCs could be disassembled as odd panels, even panels, chambers, and the final closing plate, which are exhibited in Figure 1. Except the final closing plate, the parameters of other parts were adjustable to gain the expected sound absorption performance, which consisted of the number of layers *N*, the thickness of the panel *t*_0_, the length of the side of the square aperture on the panel *a*, and the length of the chamber *T*_0_. The component parts for the SEAM–MTCs were regular in shape, and their dimensions reached the millimeter level, which indicated that they could be fabricated easily and produced in large quantities. So, many kinds of processing and manufacturing methods could be used to produce these component parts, such as wire cutting machining, laser engraving, numerical control processing technology, mold manufacturing, additive manufacturing, etc. Moreover, the materials of SEAM–MTCs, such as metal, resin, rubber, ceramic, and composite material, have larger optional ranges, provided that long as they can be processed by certain machining methods. In this study, the component parts were fabricated with different colors and different parameters using additive manufacturing equipment— the Raise 3D Pro2 printer (Shanghai Fusion Tech Co., Ltd., Shanghai, China)—based on the fused filament fabrication method. The prepared samples, made of Acrylonitrile Butadiene Styrene (ABS) resin, are shown in Figure 15. Through the combination of various component parts, the expected sound absorption performance could be obtained.

In order to examine the accuracy of the analysis of the sound absorption mechanism and principle for the SEAM–MTCs, an actual sample with four layers was fabricated, as shown in Figure 16. The selected values for this fabricated sample were as follows: the number of layers *N* was four; the thickness of panel *t*_0_ was 2 mm; the length of the side of the square aperture on the panel *a* was 5 mm; the length of the chamber *T*_0_ was 10 mm.

The fabricated sample for the SEAM–MTCs was tested using a AWA6290T tester (Hangzhou Aihua Instruments Co., Ltd., Hangzhou, China) based on the transfer function method according to the standard of GB/T 18696.2–2002 (ISO 10534–2:1998) “Acoustics–Determination of sound absorption coefficient and impedance in impedance tubes–part 2: Transfer function method” [44,45,46,47]; its schematic diagram is shown in Figure 17a. The sample was installed at the end of sample tube and held by the sample holder. The incident sound wave was generated in the sound source controlled by the noise generator and power amplifier, and the reflected sound wave was detected by two microphones fixed on the standing wave tube. The signals were preliminarily treated in the dynamic signal analyzer and further handled by the data analysis software in the workstation; then, the actual sound absorption coefficients could be derived. The distance between the two microphones was 70 mm, and the distance between microphone 2 and the surface of the sample was 170 mm, by which the actual sound absorption coefficients in the frequency range of [200 Hz, 1600 Hz] could be determined, as shown in Figure 17b. It could be intuitively judged from Figure 17b that the deviations between the simulation data and experimental data were smaller than those between the theoretical data and experimental data, which proved that the accuracy of the finite element simulation model was better than the theoretical model. This result was consistent with the former analysis of theoretical model, which could further prove that the acoustic finite element simulation chosen in this study was accurate.

The comparisons of nine resonance frequencies and the corresponding peak sound absorption coefficients in the frequency range of [200 Hz, 1600 Hz] are summarized in Table 6 for the simulation data and in Table 7 for the theoretical data; the deviations and proportions were derived by taking the experimental data as the fiducial values. It could be found that relative to the simulation data, all the actual resonance frequencies shifted slightly to the low-frequency direction, and all the actual peak sound absorption coefficients decreased slightly. The major reason for these phenomena was that there were manufacturing errors in the process to prepare the sample of SEAM–MTCs. Firstly, these component parts were prepared by additive manufacturing, which indicated that there would be expansion in the edge. Thus, relative to the ideal values in the finite element simulation, the thickness of panel and the depth of chamber would be a little larger, and the size of the aperture would be a little smaller, which resulted in the shift of the sound absorption curve to the low-frequency direction. Secondly, the parameters for each group of channels were the same in the simulation process, but the fabrication errors were unavoidable, which weakened the coupling sound absorption effect in each group of channels. On the contrary, it could be judged from Table 7 that most resonance frequencies, in theory, were smaller than the corresponding experimental data, and there existed larger deviations between peak sound absorption coefficients in theory and those in actuality. As mentioned above, there were too many assumptions, approximations, equivalences, and omissions in the theoretical modeling process, which resulted in the lower prediction accuracy for the theoretical model relative to the acoustic finite element simulation method.

## 7. Conclusions

The SEAM–MTCs was proposed and analyzed in this study. Through the structural design, parametric analysis, performance characterization, mechanism determination, and experimental validation, the main achievements are as follows.

(1)The SEAM–MTCs consisted of odd panels, even panels, chambers, and a final closing plate, and these component parts could be fabricated separately and then assembled, which could overcome two major problems for common acoustic metamaterials and metastructures. There were 11 groups of tortuous channels with different lengths, which provided a comprehensive consideration of high sound absorption performance, low manufacturing cost, and the feasibility of mass production.(2)The effects of influencing factors on the sound absorption property of the proposed SEAM–MTCs were investigated by acoustic finite element simulation, which included the number of layers *N*, the thickness of the panel *t*_0_, the size of the square aperture *a*, and the depth of the chamber *T*_0_. It could be concluded that the sound absorption curve shifted to the low-frequency direction along with the increases of *N*, *t*_0_, and *T*_0_ and the decrease of *a*; the number of layers *N* was the most important parameter.(3)The sound absorption mechanism and principle of SEAM–MTCs were investigated by the distributions of the total acoustic energy density at the resonance frequencies. The number of resonance frequency points increased from 13 to 31 with the number of layers *N* increasing from two to six, and the average sound absorption coefficient in [200 Hz, 6000 Hz] was improved from 0.5169 to 0.6160 accordingly, which was consistent with the common sound absorption principle of the Fabry–Pérot resonator. Relative to the sound absorbers in the literature [29,30] based on the same Fabry–Pérot resonance mechanism, the SEAM–MTCs could obtain a wider sound absorption band with lower manufacturing difficulty.(4)The SEAM–MTCs with four layers was fabricated and then assembled, and the experimental testing results of its sound absorption coefficients in [200 Hz, 1600 Hz] exhibited excellent consistency with simulation data, which proved the accuracy of the finite element simulation model and the reliability of analysis of the influencing factors. The deviations between the simulation data and experimental data mainly resulted from manufacturing errors, and they were smaller than those between the theoretical data and experimental data.

The proposed SEAM–MTCs had obvious advantages in adjustable sound absorption properties and convenient manufacturability, which exhibits great potential in noise control for large mechanical equipment with changing noise characteristics.

## Figures and Tables

**Figure 1 materials-16-06643-f001:**
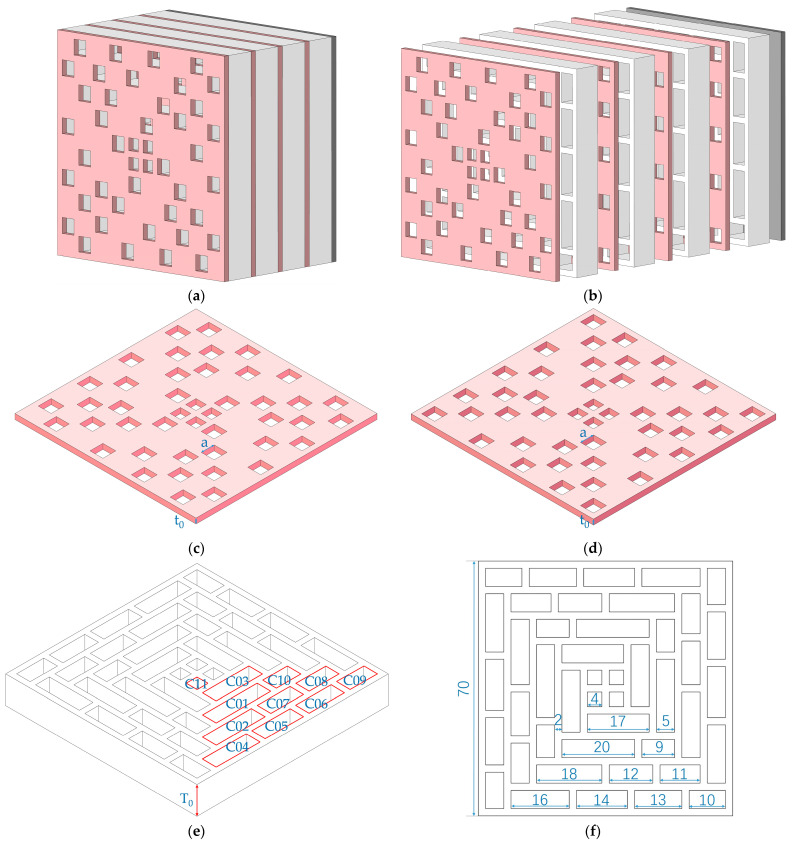
The schematic diagram of SEAM–MTCs of 4 layers. (**a**) The whole structure; (**b**) the corresponding explosion view; (**c**) the panel for odd layers; (**d**) the panel for even layers; (**e**) the chamber; and (**f**) the cross-sectional structure of the chamber.

**Figure 2 materials-16-06643-f002:**
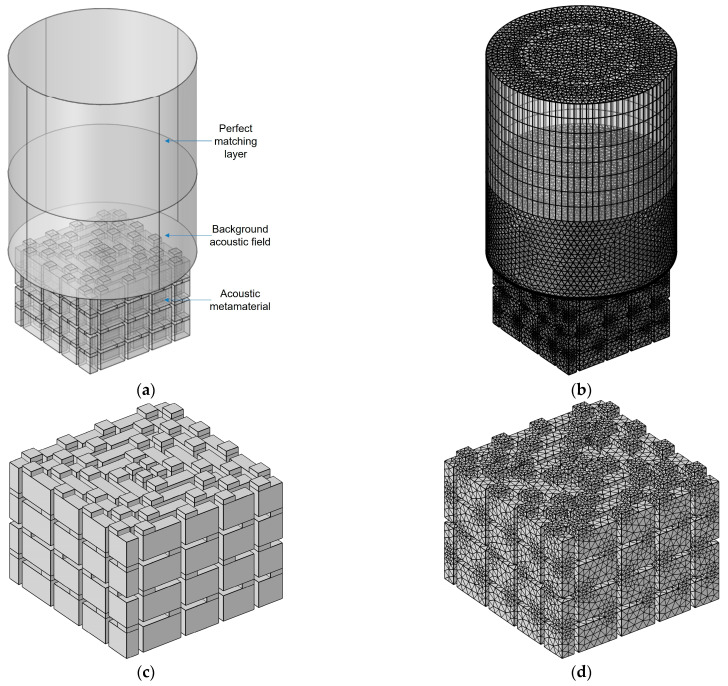
The finite element simulation model to investigate the sound absorption performance of SEAM–MTCs for 4 layers. (**a**) The whole model; (**b**) the gridded model; (**c**) the geometrical model of SEAM–MTCs; and (**d**) the gridded model of SEAM–MTCs.

**Figure 3 materials-16-06643-f003:**
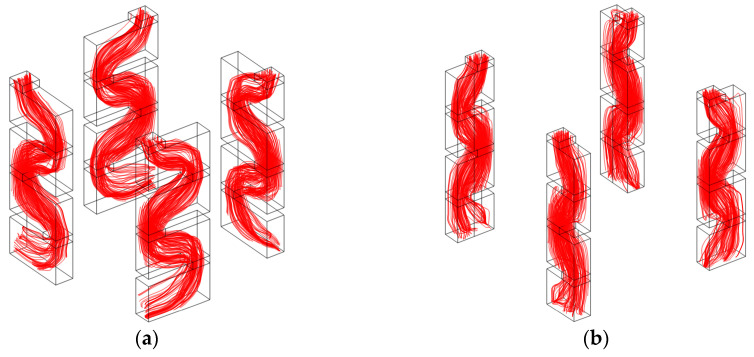
The streamline diagram of acoustic velocities in group of tortuous channels with the same parameters. (**a**) The tortuous channels when the length of the chamber is 18 mm; and (**b**) the tortuous channels when the length of chamber is 11 mm.

**Figure 4 materials-16-06643-f004:**
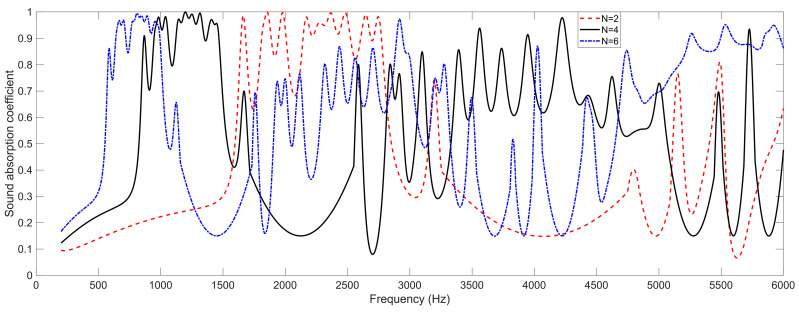
The effects of the number of layers *N* on the sound absorption property of the SEAM–MTCs.

**Figure 5 materials-16-06643-f005:**
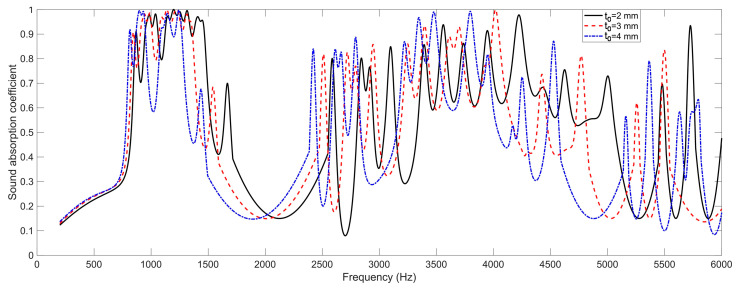
The effects of the thickness of panel *t*_0_ on the sound absorption property of the SEAM–MTCs.

**Figure 6 materials-16-06643-f006:**
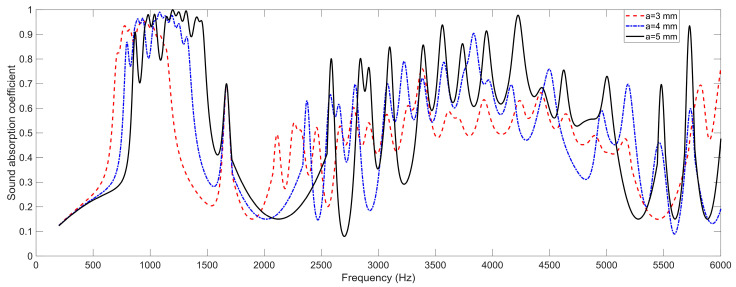
The effects of the length of the side of the square aperture *a* on the sound absorption performance of the SEAM–MTCs.

**Figure 7 materials-16-06643-f007:**
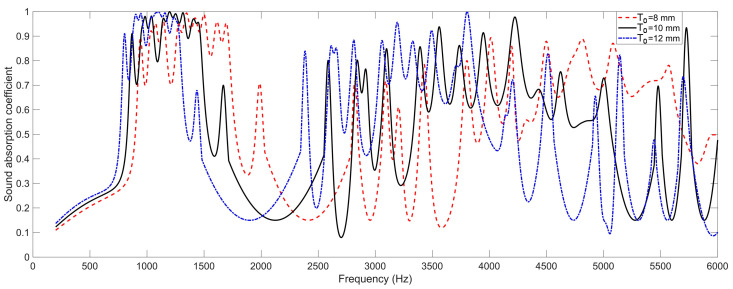
The effects of the depth of chamber *T*_0_ on the sound absorption property of the SEAM–MTCs.

**Figure 8 materials-16-06643-f008:**
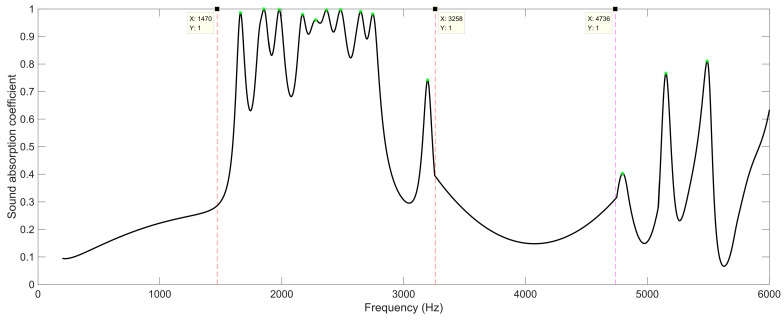
The division of sound absorption bands for the SEAM–MTCs when the number of layers was two.

**Figure 9 materials-16-06643-f009:**
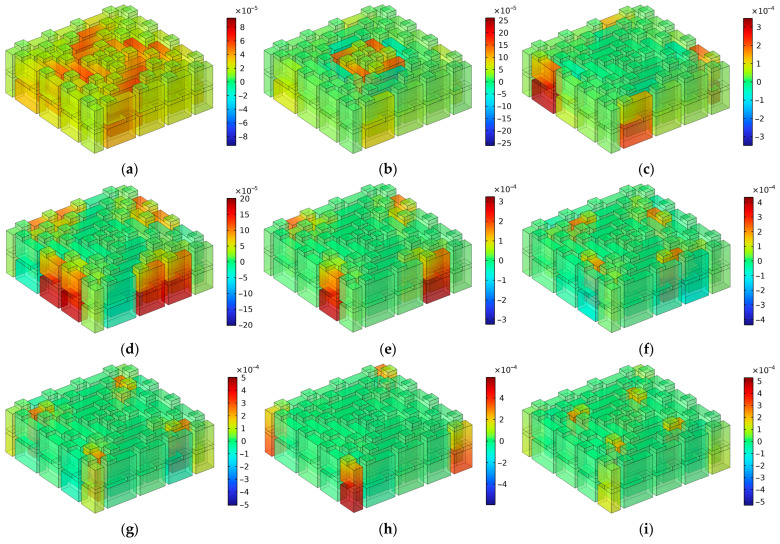
The distributions of the total acoustic energy density at the resonance frequencies when the number of layers was 2. (**a**) 1662 Hz; (**b**) 1854 Hz; (**c**) 1978 Hz; (**d**) 2170 Hz; (**e**) 2276 Hz; (**f**) 2366 Hz; (**g**) 2480 Hz; (**h**) 2646 Hz; (**i**) 2746 Hz; (**j**) 3198 Hz; (**k**) 4796 Hz; (**l**) 5150 Hz; (**m**) 5488 Hz; and (**n**) the distribution of peak sound absorption coefficients.

**Figure 10 materials-16-06643-f010:**
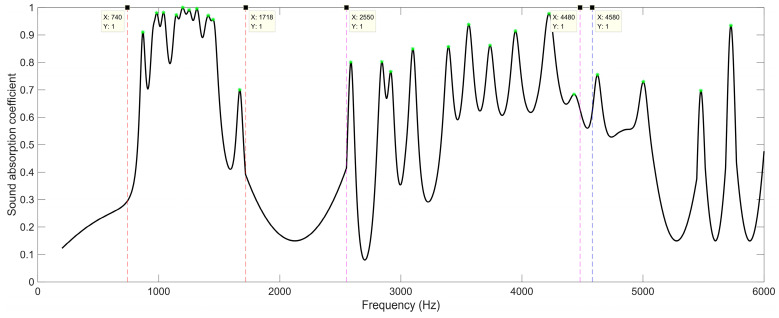
The division of sound absorption bands for the SEAM–MTCs when the number of layers was 4.

**Figure 11 materials-16-06643-f011:**

The distributions of the total acoustic energy density at the resonance frequencies when the number of layers was 4. (**a**) 868 Hz; (**b**) 982 Hz; (**c**) 1038 Hz; (**d**) 1144 Hz; (**e**) 1198 Hz; (**f**) 1252 Hz; (**g**) 1314 Hz; (**h**) 1408 Hz; (**i**) 1450 Hz; (**j**) 1670 Hz; (**k**) 2586 Hz; (**l**) 2842 Hz; (**m**) 2916 Hz; (**n**) 3098 Hz; (**o**) 3394 Hz; (**p**) 3562 Hz; (**q**) 3736 Hz; (**r**) 3946 Hz; (**s**) 4226 Hz; (**t**) 4432 Hz; (**u**) 4624 Hz; (**v**) 5004 Hz; (**w**) 5480 Hz; (**x**) 5726 Hz; and (**y**) the distribution of peak sound absorption coefficients.

**Figure 12 materials-16-06643-f012:**
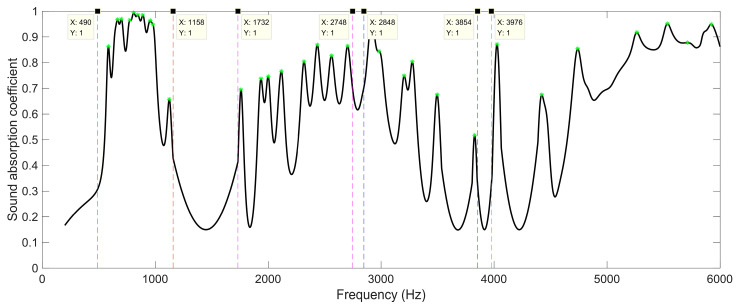
The division of the sound absorption band for the SEAM–MTCs when the number of layers was 6.

**Figure 13 materials-16-06643-f013:**
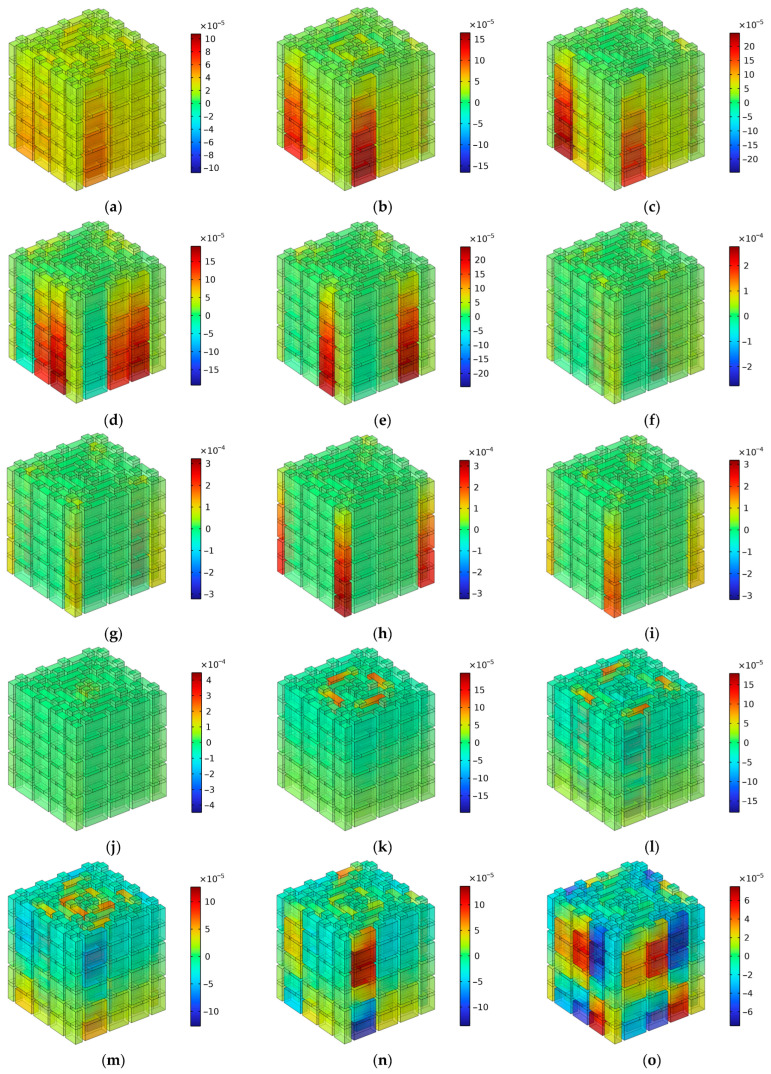
The distributions of total acoustic energy density at the resonance frequencies when the number of layers was 6. (**a**) 586 Hz; (**b**) 666 Hz; (**c**) 700 Hz; (**d**) 776 Hz; (**e**) 810 Hz; (**f**) 846 Hz; (**g**) 890 Hz; (**h**) 956 Hz; (**i**) 980 Hz; (**j**) 1124 Hz; (**k**) 1758 Hz; (**l**) 1936 Hz; (**m**) 2000 Hz; (**n**) 2116 Hz; (**o**) 2318 Hz; (**p**) 2436 Hz; (**q**) 2562 Hz; (**r**) 2704 Hz; (**s**) 2918 Hz; (**t**) 2982 Hz; (**u**) 3204 Hz; (**v**) 3276 Hz; (**w**) 3496 Hz; (**x**) 3828 Hz; (**y**) 4026 Hz; (**z**) 4422 Hz; (**aa**) 4740 Hz; (**ab**) 5264 Hz; (**ac**) 5534 Hz; (**ad**) 5712 Hz; (**ae**) 5922 Hz; and (**af**) the distribution of peak sound absorption coefficients.

**Figure 14 materials-16-06643-f014:**
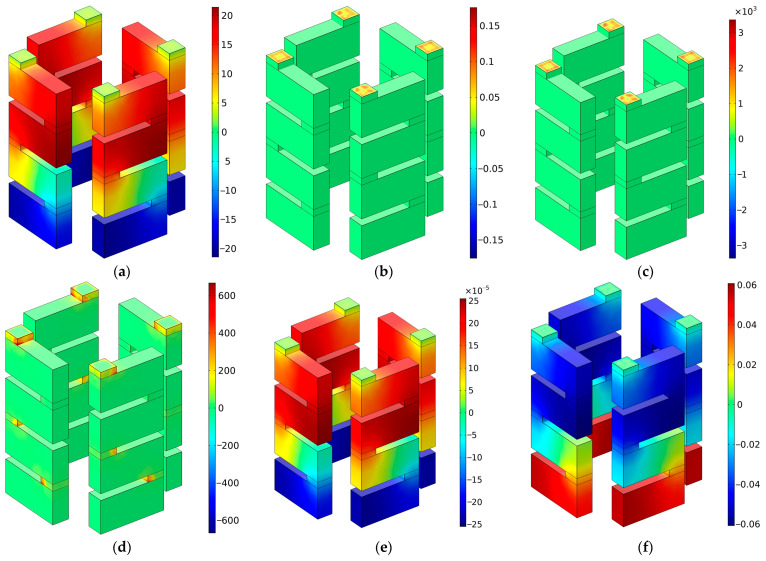
The distributions of acoustic characteristic parameters at resonance frequency of 2586 Hz when the number of layers was 4. (**a**) Acoustic pressure; (**b**) acoustic velocity; (**c**) local acceleration; (**d**) viscous power density; (**e**) total acoustic energy density; and (**f**) total specific entropy variation.

**Figure 15 materials-16-06643-f015:**
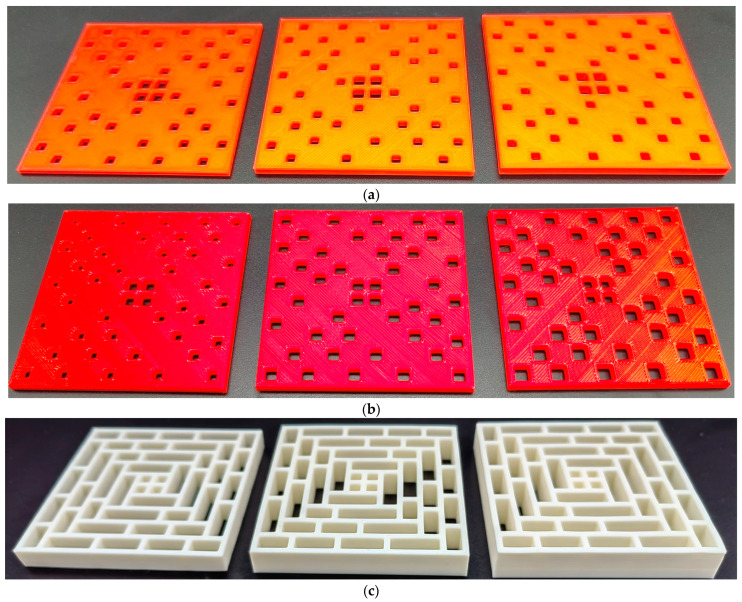
The fabricated component parts with the various parameters. (**a**) Panel with different thicknesses; (**b**) panel with different sizes of aperture; and (**c**) chamber with different depths.

**Figure 16 materials-16-06643-f016:**
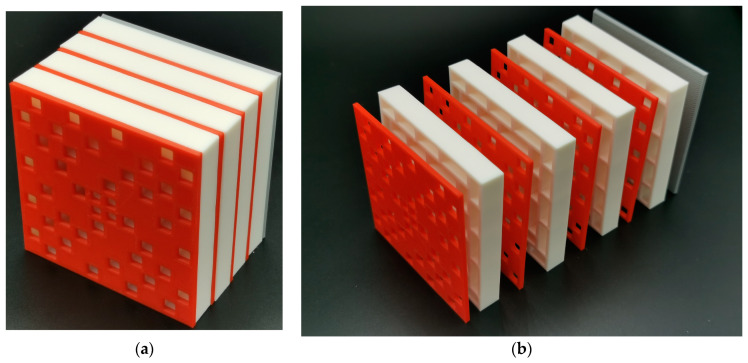
The SEAM–MTCs for 4 layers. (**a**) The assembled sample; and (**b**) the component parts.

**Figure 17 materials-16-06643-f017:**
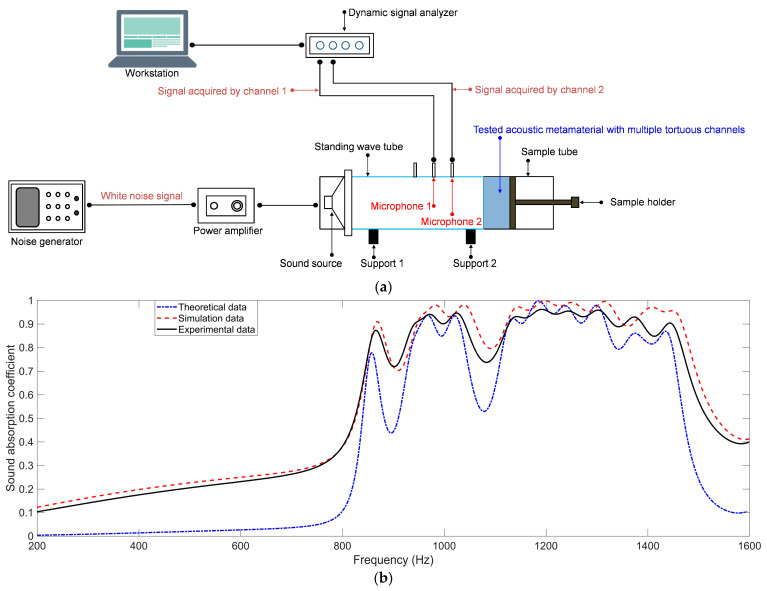
The test of the fabricated sample for SEAM–MTCs. (**a**) The schematic diagram of the testing process based on the transfer function method; and (**b**) the comparative analysis among theoretical data, simulation data, and experimental data.

**Table 1 materials-16-06643-t001:** The various parameters for the 11 groups of tortuous channels in the SEAM–MTCs.

Parameters	C01	C02	C03	C04	C05	C06	C07	C08	C09	C10	C11
Thickness of panel	*t* _0_	*t* _0_	*t* _0_	*t* _0_	*t* _0_	*t* _0_	*t* _0_	*t* _0_	*t* _0_	*t* _0_	*t* _0_
Side length of square aperture	*a*	*a*	*a*	*a*	*a*	*a*	*a*	*a*	*a*	*a*	4 mm
Depth of chamber	*T* _0_	*T* _0_	*T* _0_	*T* _0_	*T* _0_	*T* _0_	*T* _0_	*T* _0_	*T* _0_	*T* _0_	*T* _0_
Width of chamber	5 mm	5 mm	5 mm	5 mm	5 mm	5 mm	5 mm	5 mm	5 mm	5 mm	4 mm
Length of chamber	20 mm	18 mm	17 mm	16 mm	14 mm	13 mm	12 mm	11 mm	10 mm	9 mm	0 mm

**Table 2 materials-16-06643-t002:** Summary of selected parameters in the acoustic finite element simulation process.

Parameters	Value or Type	Parameters	Value or Type
The type of mesh	Extremely fine mesh	The type of acoustic field	Plane wave
The type of grid	Free tetrahedral grid	The amplitude of background field	1 Pa
The selected solver	Steady-state solver	The direction of incident wave	(0, 0, −1)
The maximum unit size	2 mm	The equilibrium pressure	1 atm
The minimum unit size	0.02 mm	The equilibrium temperature	293.15 K
The maximal unit growth rate	1.3	The number of layers in distribution	8
The curvature factor	0.2	The number of layers in boundary	8
The resolution of the narrow region	1	The stretch factor in boundary	1.2
The investigated frequency range	200–6000 Hz	The regulation factor for thickness	1

**Table 3 materials-16-06643-t003:** The correspondence of resonance frequency points to the 11 groups of channels for 2 layers.

Resonance Frequency Point	C01	C02	C03	C04	C05	C06	C07	C08	C09	C10	C11
1662 Hz	1	1	1								
1854 Hz			1								
1978 Hz				1							
2170 Hz					1	1					
2276 Hz						1					
2366 Hz							1				
2480 Hz								1			
2646 Hz									1		
2746 Hz										1	
3198 Hz											1
4796 Hz	1										
5150 Hz		1									
5488 Hz			1	1							

**Table 4 materials-16-06643-t004:** The correspondence of resonance frequency points to the 11 groups of channels for 4 layers.

Resonance Frequency Point	C01	C02	C03	C04	C05	C06	C07	C08	C09	C10	C11
868 Hz	1	1	1								
982 Hz			1	1							
1038 Hz				1							
1144 Hz					1	1					
1198 Hz						1	1				
1252 Hz							1				
1314 Hz								1			
1408 Hz									1		
1450 Hz										1	
1670 Hz											1
2586 Hz	1										
2842 Hz		1									
2916 Hz		1	1								
3098 Hz				1							
3394 Hz					1	1					
3562 Hz					1	1	1				
3736 Hz						1	1	1			
3946 Hz								1	1		
4226 Hz									1		
4432 Hz									1	1	
4624 Hz	1	1									
5004 Hz				1							
5480 Hz					1						
5726 Hz						1					

**Table 5 materials-16-06643-t005:** The correspondence of resonance frequency points to the 11 groups of channels for 6 layers.

Resonance Frequency Point	C01	C02	C03	C04	C05	C06	C07	C08	C09	C10	C11
586 Hz	1	1	1	1							
666 Hz			1	1							
700 Hz				1							
776 Hz					1	1					
810 Hz						1	1				
846 Hz							1				
890 Hz								1			
956 Hz									1		
980 Hz									1	1	
1124 Hz											1
1758 Hz	1										
1936 Hz		1									
2000 Hz		1	1								
2116 Hz			1	1							
2318 Hz					1	1					
2436 Hz					1	1	1				
2562 Hz						1	1	1			
2704 Hz								1	1		
2918 Hz	1								1	1	
2982 Hz	1									1	
3204 Hz		1									
3276 Hz			1								1
3496 Hz				1							
3828 Hz					1						
4026 Hz	1					1					
4422 Hz		1									
4740 Hz									1		
5264 Hz					1						
5534 Hz		1				1					
5712 Hz			1								1
5922 Hz	1			1			1				1

**Table 6 materials-16-06643-t006:** The comparative analysis between the simulation data and experimental data.

Resonance Frequency Point				Peak Sound Absorption Coefficient			
Simulation	Actuality	Deviation	Proportion	Simulation	Actuality	Deviation	Proportion
868 Hz	866.12 Hz	1.88 Hz	0.22%	0.9102	0.8646	0.0384	1.88%
982 Hz	972.26 Hz	9.74 Hz	1.00%	0.9805	0.9316	0.0411	1.84%
1038 Hz	1024.96 Hz	13.04 Hz	1.27%	0.9819	0.9372	0.0368	1.38%
1144 Hz	1143.55 Hz	0.45 Hz	0.04%	0.9728	0.9227	0.0425	2.03%
1198 Hz	1191.86 Hz	6.14 Hz	0.52%	0.9999	0.9526	0.0394	1.58%
1252 Hz	1244.56 Hz	5.44 Hz	0.44%	0.9916	0.9455	0.0383	1.50%
1314 Hz	1302.39 Hz	11.61 Hz	0.89%	0.9956	0.9495	0.0382	1.47%
1408 Hz	1374.13 Hz	33.87 Hz	2.46%	0.9708	0.9207	0.0424	2.04%
1450 Hz	1444.40 Hz	3.60 Hz	0.25%	0.9565	0.8951	0.0539	3.40%

**Table 7 materials-16-06643-t007:** The comparative analysis between the theoretical data and experimental data.

Resonance Frequency Point				Peak Sound Absorption Coefficient			
In theory	In actuality	Deviation	Proportion	In theory	In actuality	Deviation	Proportion
861 Hz	866.12 Hz	−5.12 Hz	−0.59%	0.7392	0.8646	−0.1254	−14.51%
970 Hz	972.26 Hz	−2.26 Hz	−0.23%	0.8899	0.9316	−0.0417	−4.48%
1024 Hz	1024.96 Hz	−0.96 Hz	−0.09%	0.8878	0.9372	−0.0494	−5.27%
1137 Hz	1143.55 Hz	−6.55 Hz	−0.57%	0.8788	0.9227	−0.0439	−4.76%
1188 Hz	1191.86 Hz	−3.86 Hz	−0.32%	0.9480	0.9526	−0.0046	−0.48%
1240 Hz	1244.56 Hz	−4.56 Hz	−0.37%	0.9283	0.9455	−0.0172	−1.82%
1302 Hz	1302.39 Hz	−0.39 Hz	−0.03%	0.9301	0.9495	−0.0194	−2.04%
1378 Hz	1374.13 Hz	3.87 Hz	0.28%	0.8169	0.9207	−0.1038	−11.27%
1438 Hz	1444.40 Hz	−6.4 Hz	−0.44%	0.8260	0.8951	−0.0691	−7.72%

## Data Availability

The data that support the findings of this study are available from the corresponding author upon reasonable request.

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
