# Peer review of "Analysis of Influencing Factors for Stackable and Expandable Acoustic Metamaterial with Multiple Tortuous Channels"

_materials, 2023, doi:10.3390/ma16206643_

Round 1

Reviewer 1 Report

The article discusses the parameters and properties of acoustic metamaterial.

A complex and interesting research approach was used.

It was shown that the analyzed acoustic metamaterial is capable of effectively absorbing sound.

The metamaterial has been proven to have noise control potential for large mechanical devices with changing noise characteristics.

The topic is original and justified in the indicated field.

So far, no similar variant of the analysis has been carried out in this regard.

The methodology is clearly described.

The conclusions are consistent with the evidence and arguments presented. The conclusions correspond to the main task set.

The list of references is exhaustive.

Tables and figures are clear and legible.

Author Response

Response to reviewer’s comments

General Comment:

The article discusses the parameters and properties of acoustic metamaterial.

A complex and interesting research approach was used.

It was shown that the analyzed acoustic metamaterial is capable of effectively absorbing sound.

The metamaterial has been proven to have noise control potential for large mechanical devices with changing noise characteristics.

The topic is original and justified in the indicated field.

So far, no similar variant of the analysis has been carried out in this regard.

The methodology is clearly described.

The conclusions are consistent with the evidence and arguments presented. The conclusions correspond to the main task set.

The list of references is exhaustive.

Tables and figures are clear and legible.

Response:

Thank you very much for your helpful review and positive assessment to our manuscript. In this research, the stackable and expandable acoustic metamaterial with the multiple tortuous channels was developed to reduce the noise generated by large mechanical equipment. The number of resonance frequencies increased from 13 to 31 along with the number of layers N increasing from 2 to 6, and the corresponding average sound absorption coefficients in [200 Hz, 6000 Hz] was significantly improved from 0.5169 to 0.6160. The experimental validation of actual sound absorption coefficients in [200 Hz, 1600 Hz] showed excellent consistency to simulation data, which proved the accuracy of acoustic finite element simulation model and the reliability of analysis on the influencing factors. This stackable and expandable acoustic metamaterial had great potential in the field of noise reduction. Meanwhile, We have revised the whole manuscript carefully according to your and other reviewers’ comments, and these corrections are highlighted in yellow in the revised manuscript.

Reviewer 2 Report

The authors present review paper on the use of stackable material composed of multiple individual layers. The layers have rectangular channels that form several acoustic cavities when put together. In general, I find the topic very interesting from a fundamental point of view and extremely useful for some industrial applications.

The introduction reads as a disconnected review of the literature. The authors should try to improve the readability of the storyline, which is hard to follow in this way. The authors should educate the non-specialized reader on how the content is important or novel.  

The theory is not discussed at all, the authors just say that do not agree with the experiment without any quantitative evaluation.

I could not find any explicit reference to the numerical model employed. Did the authors use a commercial software? In that case they should specify with packages or modules were employed and detail the boundary conditions to allow the readers to reproduce the work. In the case that the analysis was made with an in-house written code, it would be expected that the authors can share it in a repository.

The experimental part lack of any detail on the equipment used and many important details on the experimental method. Everything is written in generic as these details would not matter. I could not even find explicitly the material used to fabricate the layers. This is not acceptable.  

The topic treated by this paper is relevant, but the poor quality of English prevents the reader to enjoy and get the most of this work. Furthermore, the absence of critical details on the method along with some other mayor issues on the scientific part prevents me to recommend this manuscript on its current state. The authors should dramatically improve its quality and address all my comments to be considered for publication in Materials.

One major point to criticize, is the way the paper is written. Some phrases are put in an odd way or are difficult to understand. I would recommend to the authors to improve the wording of the paper with an English native speaking colleague or hiring an editorial service. The readers would appreciate that and the message behind the research will be delivered in a more effective way. Some phases are extremely long and impossible to follow and some names like: “stackable and expandable acoustic metamaterial with multiple tortuous channels” are repeated too many times.

Author Response

Response to reviewer’s comments

General Comment: The authors present review paper on the use of stackable material composed of multiple individual layers. The layers have rectangular channels that form several acoustic cavities when put together. In general, I find the topic very interesting from a fundamental point of view and extremely useful for some industrial applications.

Response:

Thank you very much for your kind review to our manuscript and constructive suggestion to our study. We have revised and corrected the whole manuscript carefully according to your and the other reviewers’ comments. The responses to your helpful comments are as follows.

  1. The introduction reads as a disconnected review of the literature. The authors should try to improve the readability of the storyline, which is hard to follow in this way. The authors should educate the non-specialized reader on how the content is important or novel.

Response:

Thank you very much for your valuable comment and helpful suggestion. The consistency of presentation on the review of literatures in introduction section is improved by adding some conjunctions and some summary descriptions. Meanwhile, the importance of this research and the novelty of this manuscript are pointed out in the introduction section, which aims to make the manuscript more readable and easier to understand.

  1. The theory is not discussed at all, the authors just say that do not agree with the experiment without any quantitative evaluation.

Response:

Thank you very much for your meaningful question. The theoretical data of the proposed acoustic metamaterial with 4 layers is compared with simulation data and experimental data in the Figure 17b in the revised manuscript, which can visually show differences among the 3 kinds of results. Meanwhile, taking the experimental data as the criterion, the deviations for theoretical data and those for simulation data are derived, as shown in Table 6 in the revised manuscript, which can further give quantitative evaluation to prove that prediction accuracy of the acoustic finite element simulation is better than that of theoretical model. These corresponding modifications and corrections are highlighted in yellow in the revised manuscript.

  1. I could not find any explicit reference to the numerical model employed. Did the authors use a commercial software? In that case they should specify with packages or modules were employed and detail the boundary conditions to allow the readers to reproduce the work. In the case that the analysis was made with an in-house written code, it would be expected that the authors can share it in a repository.

Response:

Thank you very much for your worthy question. The COMSOL 5.5 Multiphysics field simulation software is used to build the acoustic finite element simulation model in this research. We apologize for not pointing this out in the manuscript, and the simulation platform is added in the revised manuscript. Meanwhile, the major parameter setting in the simulation process is summarized in the Table 2, which can allow the readers to reproduce the results shown in this manuscript.

  1. The experimental part lack of any detail on the equipment used and many important details on the experimental method. Everything is written in generic as these details would not matter. I could not even find explicitly the material used to fabricate the layers. This is not acceptable.

Response:

Thank you very much for your valuable question. The fabricated sample for the SEAM-MTCs is tested by the AWA6290T tester according to transfer function method, and the detailed information is added in the section “6. Experimental Validation” in revised manuscript, such as manufacturer, reference standard, etc. Meanwhile, the sample in this research is fabricated by the Raise 3D Pro2 additive manufacturing equipment with ABS resin, and the detailed information is added as well. We apologize for not pointing these out in the manuscript.

  1. The topic treated by this paper is relevant, but the poor quality of English prevents the reader to enjoy and get the most of this work. Furthermore, the absence of critical details on the method along with some other mayor issues on the scientific part prevents me to recommend this manuscript on its current state. The authors should dramatically improve its quality and address all my comments to be considered for publication in Materials.

Response:

Thank you very much for your kindly suggestion and patient review. The whole manuscript is polished to improve the quality of English. We have revised the manuscript thoroughly according to your and the other reviewers’ constructive comments. Some essential information is added, and some figures and tables are adjusted, which are highlighted in yellow in the revised manuscript.

  1. One major point to criticize, is the way the paper is written. Some phrases are put in an odd way or are difficult to understand. I would recommend to the authors to improve the wording of the paper with an English native speaking colleague or hiring an editorial service. The readers would appreciate that and the message behind the research will be delivered in a more effective way. Some phases are extremely long and impossible to follow and some names like: “stackable and expandable acoustic metamaterial with multiple tortuous channels” are repeated too many times.

Response:

Thank you very much for your meaningful and helpful suggestion. The wording of this manuscript is polished by ourselves and further improved by the English native speaker in the field of acoustic metamaterial for the noise reduction. In particular, the presentation “stackable and expandable acoustic metamaterial with multiple tortuous channels” is abbreviated as “SEAM-MTCs” in the revised manuscript after first appearance, which aims to avoid this kind of long phases. We apologize for the unsatisfying presentation in this manuscript, and we will attempt to improve our writing ability in the future research.

Reviewer 3 Report

In this manuscript, the authors developed a novel stackable and expandable acoustic metamaterial which can reduce noise generated by large mechanical equipment. The influences of the number of layers, thickness of the panel, size of square aperture and the depth of chamber on the sound absorption performance of the proposed acoustic metamaterial were investigated. After the simulation, the reliability of the acoustic metamaterial was validated by experimental test as well. Based on the results of the study, the metamaterial developed here have the potential to be used in various application areas such as small-scale electroacoustic devices and sensors to reduce or control noise. In general, the topic is interesting, and the results are valuable. But before publishing but I believe that there are some points that need to be improved.

- Explain in more detail the metamaterial fabrication, its production method and the materials used. Materials and Design section must be improved.

- Apart from noise reduction, have other acoustic properties of the acoustic metamaterial been investigated, such as reflection coefficient etc.? If there is an advantage or disadvantage in other acoustic properties compared to other acoustic materials commonly used in the literature, please mention it in the manuscript.

Author Response

Response to reviewer’s comments

General Comment:

In this manuscript, the authors developed a novel stackable and expandable acoustic metamaterial which can reduce noise generated by large mechanical equipment. The influences of the number of layers, thickness of the panel, size of square aperture and the depth of chamber on the sound absorption performance of the proposed acoustic metamaterial were investigated. After the simulation, the reliability of the acoustic metamaterial was validated by experimental test as well. Based on the results of the study, the metamaterial developed here have the potential to be used in various application areas such as small-scale electroacoustic devices and sensors to reduce or control noise. In general, the topic is interesting, and the results are valuable. But before publishing but I believe that there are some points that need to be improved.

Response:

Thank you very much for your kind review to our manuscript and constructive suggestions to our research. We have revised the whole manuscript carefully according to your and the other reviewers’ comments, and these corrections are highlighted in yellow in the revised manuscript. The response to each your comment is as follows.

  1. Explain in more detail the metamaterial fabrication, its production method and the materials used. Materials and Design section must be improved.

Response:

Thank you very much for your worthy suggestion. The detailed information on the metamaterial fabrication is added in the section “2. Materials and Design”, which consists of the production method, the used material, the other available method, and the other optional materials, and these corrections are highlighted in yellow in the revised manuscript.

  1. Apart from noise reduction, have other acoustic properties of the acoustic metamaterial been investigated, such as reflection coefficient etc.? If there is an advantage or disadvantage in other acoustic properties compared to other acoustic materials commonly used in the literature, please mention it in the manuscript.

Response:

Thank you very much for your valuable comment. In the initial manuscript, only the sound absorption coefficients of this proposed acoustic metamaterial are studied. According to your meaningful suggestion, we have added discussions at the end of the section “5. Sound Absorption Mechanism” as new subsection “5.4 Acoustic characteristic parameters”. In this subsection, the distributions of acoustic pressure, acoustic velocity, viscous power density, and total specific entropy variation are discussed at the certain resonance frequency, which can better reveal the sound absorption mechanism and characters of this proposed acoustic metamaterial. Meanwhile, the sound reflection coefficient has direct relationship with sound absorption coefficient, as shown in the following figure.

The total acoustic energy Eo consists of the reflection energy Er, the absorption energy Ea, and transmission energy Ei. The absorption coefficient α, reflection coefficient γ and transmission coefficient β can be calculated by the following equations.

α=1-γ=( Ea+ Ei)/ Eo

γ= Er/ Eo

β= Ei/ Eo

In the experimental testing process, the sample is fixed in the sample tube and held by the sample holder, which are made of metal with the large thicknesses, so transmission coefficient β can be considered as 0. The relationship between absorption coefficient α and reflection coefficient γ is α+γ=1. So, the reflection coefficient γ and transmission coefficient β are not discussed in this research.

Moreover, sound absorption property of the proposed acoustic metamaterial is compared with the similar sound absorbing materials or structures under the same conditions, which is added in the end of section “4. Parametric Analysis” in the revised manuscript and the modifications are highlighted in yellow.
